# Benefits and Pitfalls of HPLC Coupled to Diode-Array, Charged Aerosol, and Coulometric Detections: Effect of Detection on Screening of Bioactive Compounds in Apples

**DOI:** 10.3390/molecules26113246

**Published:** 2021-05-28

**Authors:** Marcela Hollá, Aneta Bílková, Pavel Jakubec, Stanislava Košková, Hana Kočová Vlčková, Dalibor Šatínský, František Švec, Hana Sklenářová

**Affiliations:** 1Department of Analytical Chemistry, Faculty of Pharmacy in Hradec Králové, Charles University, 50005 Hradec Králové, Czech Republic; hollama@faf.cuni.cz (M.H.); Aneta.BILKOVA@vsuo.cz (A.B.); jakubep1@faf.cuni.cz (P.J.); stanislava.koskova@tul.cz (S.K.); VLCKH3AA@faf.cuni.cz (H.K.V.); satinsky@faf.cuni.cz (D.Š.); svecfr@faf.cuni.cz (F.Š.); 2Research and Breeding Institute of Pomology Holovousy Ltd., 50801 Hořice, Czech Republic

**Keywords:** apple extracts, complex matrices, detection sensitivity, antioxidants, phenolic compounds, diode-array detection, charged aerosol detection, coulometric detection

## Abstract

The new screening method for rapid evaluation of major phenolic compounds in apples has been developed. Suitability of coupling HPLC/UHPLC separation with the diode-array detection and universal charged aerosol detection with respect to the presence of interfering substances was tested. Characteristics of both detection techniques were compared and method linearity, limits of detection and quantitation, and selectivity of them determined. Student t-test based on slopes of calibration plots was applied for the detailed comparison. The diode-array detection provided the best results regarding sensitivity and selectivity of the developed method in terms of evaluation of phenolics profiles. The response of the charged aerosol detector was negatively affected by co-eluting substances during rapid-screening analyses. Coulometric detection was used for advanced characterization of extracts in terms of antioxidant content and strength to obtain more complex information concerning sample composition. This detection also allowed evaluation of unidentified compounds with antioxidant activity. HPLC/UHPLC separation using a combination of diode-array and coulometric detectors thus represented the best approach enabling quick, yet complex characterization of bioactive compounds in apples.

## 1. Introduction

Interfering sample components are the common problem in analyses of complex matrices. The presence of the sample matrix and its potential co-elution with analytes can affect detection response. This problem is well known in the field of mass spectrometry (MS) as matrix effects mainly while applying the electrospray ionization (ESI). The matrix alters the signal of the analyzed components causing overestimation or underestimation of their content and results in poor analytical accuracy, reproducibility, and linearity [1].

Matrix effects can be evaluated by comparison of calibration slopes of matrix-matched and standard calibration curves [2]. Variety of approaches have been developed to reduce or eliminate matrix effects. Extensive cleanup methods [3,4], efficient chromatographic separation [2,5,6], investigation of peak purity [7], and sample dilution have been reported [3,8,9]. Use of isotope labelled internal standards is widely applied in MS. Nevertheless, these standards are usually expensive and not always available [10,11]. Selection of the suitable sample preparation method must be closely related to the properties of the sample. Its type, amount, composition, physicochemical properties, and concentration of target analytes should be considered.

The impact of interfering components on quantitation of analytes in complex matrices using MS and other detection techniques was compared in several previous works. Analyses of environmental [2,10,11], plant [9,12,13,14], and biological samples [15,16] are typically affected by a multi-component sample matrix. For example, Verdu et al. compared determination of 15 major phenolic compounds in apple ciders using UHPLC-UV and UHPLC-MS/MS. Although they found an excellent correlation between both methods for most of the analytes, the quantitation of five compounds was affected by interfering components. Using diode-array detection (DAD), the content of chlorogenic acid was overestimated, while quercitrin, epicatechin, 4-p-coumaroylquinic acid, and avicularin were underestimated when compared to MS quantitation. These authors concluded that LC-MS determination could not be ubiquitously applied to the linear quantification of all phenolic compounds, especially when dealing with complex plant matrices [12]. Similarly, quantification of oxamyl pesticide [17], and phenolic compounds in wine [18] when applying DAD compared to MS with selected reaction monitoring was influenced by co-eluting matrix components. Although DAD is widely applied for analyses of complex matrices such as plant extracts, current literature lacks its comparison with other types of detection such as electrochemical, fluorescence, and universal detection with respect to interfering components.

A significant part of current reports evaluates contents of health-promoting substances with antioxidant properties in a variety of plants [19,20]. For example, the complexity of compounds contained in apples is exemplified by polysaccharides, phytosterols, proteins, vitamins, and trace elements. This collection of compounds along with the wide range and similarity of phenolic structures augments requirements in chromatographic analyses concerning column efficiency, detection selectivity, and sensitivity [21].

Spectrophotometric DAD detection is easily applicable due to the wide-spread presence of absorbing chromophores in analyte molecules [22,23]. As the single detection technique, it is widely used for the evaluation of phenolic compounds in fruit, vegetables, and medicinal herbs [24,25,26,27,28]. Acquisition of full UV-VIS spectrum in the format of a 3D plot allows evaluation of peak purity, reveals possible peak co-elutions, and enables collection of spectral libraries. DAD can also be coupled with another detector featuring different detection concept. Combinations with electrochemical [29], fluorescence [30], and MS detectors are frequent [12,31,32,33,34]. Due to programmable excitation and emission wavelengths, fluorescence detection (FLD) can be used for sensitive and selective evaluation of gallic acid, catechin, and epicatechin [7,30,35].

The connection of DAD and FLD for quantitation of phenolic compounds appears to be advantageous since it allows excellent peak identification and purity evaluation via DAD with an additional peak confirmation using FLD. A combination of both techniques was used by Viñas et al. for quantitation of polyphenols in wine [7]. Identification was achieved via comparison of retention times with standards, absorbance ratio at two wavelengths, and by comparison with FLD. Evaluation of peak purity was assessed from the similarity between spectra at peak apex and at each point of that peak, and by a ratio of chromatograms monitored at two different wavelengths. Combination of proper peak identification and confirmation of peak purity allowed to diminish influence of interfering components [7].

Non-spectral and universal detection techniques, such as charged aerosol detection (CAD) have also been used [29,31,36]. The main advantage of CAD is the quantitation of all non-volatile analytes including even poorly UV absorbing compounds and compounds that do not have any chromophores. However, to enhance the selectivity of analyses, this approach often required use of long separation times [29,31]. Its general disadvantages such as a narrow linear range and the absence of spectral information can be solved via hyphenation with DAD [37,38,39].

Electrochemical detection such as coulometric detection (CD) and cyclic voltammetry is another suitable tool to determine the overall antioxidant capacity of complex plant samples [29,40,41,42]. CD is very sensitive to electroactive substances and enables determination of all oxidizable substances. Each compound that can be oxidized on the detector electrodes provides a response that represents additional information concerning substances that are potential antioxidants even if present only at low levels. For in vitro spectrophotometric evaluation of antioxidant activity, also use of ABTS, DPPH, FRAP, and ORAC tests are commonly accepted [43,44,45]. The major components with strong antioxidant activity can be easily identified by chromatographic analysis of plant extract before and after treatment with reactive components of the above-named tests [43]. For additional nutritional characterization of samples and structural characterization of isolated compounds nuclear magnetic resonance can be employed [43,44,45,46].

In this report, we focus on simple spectral and electrochemical detection techniques instead of highly sensitive but less affordable mass spectrometry [47]. We discuss the suitability of DAD and CAD detection techniques for identification and quantitation of selected analytes in complex apple matrices. The overall biological activity of apples is also assessed using CD. The comparison was focused on detection techniques instead of on the analytical method while quick LC separation is applied in connection with both DAD and CAD. The effect of interfering components was studied with a wide range of cultivars and the main problems of cultivars with higher chlorogenic acid content are pointed out. The comparison of different detections focused on data that could be generalized.

## 2. Results and Discussion

### 2.1. HPLC Method Development

Method optimization followed our procedure published elsewhere [48] with the following modifications: the previous LC method was based on application of core shell C18 stationary phase that was changed to fully porous C18 phase with modification for higher resolution of polar substances. Then, gradient profile was modified to increase resolution of substances close to chlorogenic acid. The Luna Omega Polar column retained even highly polar gallic acid and prevented its loss. This column enabled the desired separation of analytes of interest and produced symmetrical peaks. Examples of apple extract separation monitored using DAD, CAD, and CD are shown in Appendix A.

### 2.2. HPLC Method Validation

Our HPLC method was validated with respect to the European Medicines Agency (EMA) validation guideline [49] for the determination of seven selected plant phenolic compounds in methanolic apple extracts to confirm the reliability of the results. The EMA guidelines are very close to the Food and Drug Administration counterpart that is intended for application to the food matrix but more specific in some validation parameters. Thus, they were applied to this work. Additionally, EMA validation guidelines describe quite well cross validation of a method, which is our case. The performance parameters including system suitability test, linearity, quantitation limits, detection limits, accuracy, precision, and selectivity are referred to in the following sections. Due to the focus of this study on comparison of detection techniques, the emphasis was placed on the major detection parameters, i.e., linearity, limits of detection and quantitation, and selectivity.

#### 2.2.1. System Suitability

System suitability was evaluated with six replicate injections of 5 µg/mL mixed standard solution and characteristics comprising repeatability of the retention time, peak area, symmetry factor, resolution of the peaks, peak capacity, and retention factor were determined. The results of the system suitability test are summarized in Table 1. Repeatability of retention time and peak area was calculated as the relative standard deviation (RSD) of six replicated injections of mixed standard solution. The repeatability of retention time and peak area maintained RSD values at less than 1.0% (n = 6). Peak symmetry ranged for DAD from 0.82 to 1.20 and for CAD from 0.86 to 1.16. All peaks were well resolved with the resolution of at least 7.12 (chlorogenic acid, DAD 280 nm). The resolution of gallic acid represented by the first peak past the dead volume was always higher than 5. All these parameters met the acceptance criteria of EMA.

#### 2.2.2. Linearity

The linearity of the proposed method was determined by DAD and CAD for all analyzed compounds in a concentration range of 0.10–20 µg/mL. This concentration range was selected according to the expected content of analytes in apple extracts. Each point of the peak area vs. concentration calibration plot was measured in triplicate. The obtained results were affected by small relative values of the peak areas that were evaluated using the mentioned instrumentation. The precision increased using evaluation of all measurements. Linear regression using the least-squares method was used to draw the calibration plot. DAD featured the widest linear range of 0.10–20 µg/mL for gallic acid, chlorogenic acid, phloridzin, quercetin, and phloretin. A range of 0.25–20 µg/mL was determined for epicatechin and rutin. The lower limits of the linear range obtained by CAD were 4–10 times higher compared to DAD. According to the values of coefficients determined from the calibration plots, acceptable linearity (R^2^ ≥ 0.99) was typical for all the compounds.

Results including calibration ranges, regression equations, and determination coefficients are summarized in Table 2. The variance between detectors is primarily caused by different operating principles of the detectors, i.e., universal CAD and specific DAD that detects substances containing chromophores enabling absorption of spectral wavelengths. The variance within detector can be caused by different structures of analytes causing different ability to absorb certain spectral wavelength (DAD) and formation of non-volatile analytes after evaporation of mobile phase (CAD). The selectivity was again enhanced, and peak purity could be evaluated in the case of DAD. Using CAD, co-elution with other substances cannot be visualized.

#### 2.2.3. Detection and Quantitation Limits

The limits of detection (LOD) and quantitation (LOQ) were calculated from the calibrations. The lowest point at the calibration curve was accepted as LOQ being equal to a signal-to-noise ratio of 10. LOD was represented with a signal-to-noise ratio of 3. The LOQ values in DAD ranged from 0.1 to 0.25 µg/mL while for CAD were 2–10 times higher, ranging from 0.50 to 1.0 µg/mL. The LOD and LOQ values are included in Table 2.

Our LOD and LOQ values using CAD detection compare favorably to those reported by Plaza et al. [29]. For example, our LOD value for chlorogenic acid is almost 20 times lower and for rutin, phloridzin, and quercetin 4–10 times lower. These improvements likely originate from a technical adjustment of our HPLC/UHPLC system. We reduced the extra-column volumes via use of capillaries with narrower diameter that decreased the peak broadening thus contributing to enhancements in LOD and LOQ limits for CAD detection.

#### 2.2.4. Accuracy, Precision, and Selectivity

Accuracy of the developed method expressed in terms of compound recovery at six concentration levels of 0.2, 1, 5, 10, 15, and 20 µg/mL is summarized in Table 3. The values indicate sufficient recoveries for all analytes in a range of 85.6–116.0% for DAD, and of 86.7–103.5% for CAD with the RSD 0.6–4.2% and 0.6–4.0% for DAD and CAD, respectively.

The precision of our method, also shown in Table 3, was expressed as the closeness between a series of measurements obtained from multiple sample preparation under defined conditions. Six repeated preparations were tested in a single day and evaluated in terms of the intra-day precision. The RSD values for the peak areas of analytes expressed as the repeatability of extract preparation ranged from 1.9 to 7.1% for DAD and from 5.1 to 7.8% for CAD.

For obvious reasons, the blank apple extract matrix cannot be obtained for the evaluation of DAD and CAD selectivity. Thus, a high priority was put on proper peak integration. The valley-to-valley integration method was used, and the extracted peak spectra were compared with those of standards. Additionally, the purity of integrated peaks was verified using peak purity and match factor function of the software. Phenolic compounds were successfully separated within 8 min and all the peaks were free of interferences coinciding with the retention times of the analytes. The selectivity was confirmed with the absence of interferences detected at retention times corresponding to target analytes as demonstrated with comparing chromatogram of the spiked sample and pure extraction solution injection. The results verified that our method had sufficient selectivity for the determination of selected phenolic compounds in apple extracts.

### 2.3. Comparison of Detection Techniques Applied to Apple Extracts Analysis

#### 2.3.1. Sensitivity

Comparison based on slopes of regression curves that could be easily carried out enabled the estimation of sensitivity of our detection techniques. Student *t*-test was applied for statistical evaluation of the difference. Table 4 compares the sensitivities of DAD and CAD for all seven phenolics. The difference was confirmed by two-tailed *t*-test with corresponding *p*-value calculation. For example, the difference in sensitivity for epicatechin was the smallest among the tested analytes (*p* = 0.0000258). Therefore, we assume that all the differences across all tested analytes are statistically significant at a significance level α of 0.05.

#### 2.3.2. HPLC Determination of Selected Bioactive Compounds

Our present work aimed at the consequence of interfering components on detection techniques while enabling rapid and reliable evaluation of selected phenolic compounds in apple extracts. Therefore, we attempted the improvements in selectivity and sensitivity of DAD and CAD detection. Comparable studies [24,25,26,27,28] applying only DAD for the evaluation of selected phenolic compounds lacked information regarding antioxidant activities of single compounds. Similarly, CAD alone does not enable compound identification and had to be combined with other detection techniques [29,31,36]. We applied LC with high-resolution mass spectrometry (HRMS) to confirm the identity of the quantified analytes (see Appendix A).

Results of phenolic profiles presented in Appendix A were compared following the EMA validation guideline that recommends the mean accuracy of the results obtained by cross-validated methods to be within 15–20%. Total polyphenol content was based on the sum of the detected main polyphenolic compounds. SD values are included in Appendix A with the phenolic content results. The highest RSD was applied to calculate SD for total content of polyphenols. Based on the comparison of phenolic profiles evaluated applying both detection techniques and presented in Appendix A, unilateral overestimation or underestimation of quantified compounds was observed. The differences were analyte specific.

For example, differences ranging from −6.73 to +9.68% in the levels of quantified chlorogenic acid fell within recommended 15% accuracy limit. Compared to CAD, DAD at 320 nm had sufficient sensitivity and selectivity to quantify chlorogenic acid also at low levels in cultivars ‘Meteor’, ‘Rubinstep’, ‘Golida’, ‘Reluga’, and ‘Topaz’ (0.35–1.18 µg/mL) no matter of the presence of the complex sample matrix. No response or response under LOQ was observed for the respective cultivars using CAD. This can be explained by the lack of detection sensitivity to monitor the signal of chlorogenic acid that was overlapped by a signal of minor co-eluting substances. The within 20% accuracy limit condition of epicatechin quantitation was met only for the ‘Rubinola’ cultivar (−2.04%). In general, overestimation of epicatechin was typical for CAD compared to DAD with a range from −25.85% to +1350.65%. A similar tendency to overestimate results by CAD was found for rutin, where results were overestimated from +5.11% to +464.1% compared to DAD. On the other hand, lower concentrations quantified by DAD at 280 nm (0.25–0.96 µg/mL) were not quantified by CAD. Quantitation of phloridzin, which is a flavonoid typically present in apples, was more sensitive using DAD at 280 nm although this compound was present only at moderate levels. The ability to quantify low concentration of a compound by DAD was far more pronounced for quercetin that was quantified in 9 cultivars by DAD at 365 nm but in none by CAD. When results obtained by one of the detectors were close to LOQ, significant differences were observed.

Similar amounts of total polyphenols quantified using DAD and CAD detection were found for cultivars ‘Angold’, ‘Artiga’, ‘Golden Delicious’, ‘Meteor’, and ‘Rubinola’ despite the mentioned detector-based differences in determination of individual quantified polyphenols. It should be pointed out that such a correlation can be caused by analyte specific inclination of the detector such as DAD to chlorogenic acid and CAD to epicatechin. This phenomenon can result in similarity in total polyphenols content obtained by both methods. Overall, DAD allowed quantitation of analytes present also at lower concentrations. In contrast, the use of CAD for fast screening analysis of complex apple extracts featured both diminished sensitivity and selectivity. The analytes were not detected or were under the LOQ because their signal was overlapped by a co-eluted sample matrix. Moreover, when the analytes were present at higher concentrations, CAD was prone to overestimate them due to the signal of the co-eluting matrix.

Our results prove that CAD is more sensitive to influence of interfering components than DAD when applied for rapid analyses of the complex samples. The main reason appears to be the combination of insufficient resolution of compounds and lower selectivity of the universal CAD detection. These factors lead to the overlapping of signals of individual components and thus to their overestimation. These findings are similar to those reported in other studies using complex samples (vide infra) [29,31,50,51]. Thus, specific attention must be paid to the quantification of complex samples while using CAD. Our approach compares favorably with similar studies using CAD for quantitation of complex samples that used much longer mobile phase gradient resulting in extended elution times. For example, Baker et al. used 125 min long chromatographic separation time with DAD and CAD detection for quantitation together with HRMS for identification and characterization of constituents in standardized botanical *Ginkgo biloba* leaf extract [50]. Despite the long-lasting separation, they observed a co-elution in some CAD peaks as confirmed by HRMS. Plaza et al. applied 40 min gradient elution plus 10 min equilibration step for the separation of phenolic compounds in apple extracts [29]. The gradient of a similar length was also applied by Granica et al. for quantitative characterization of polyphenols in a herb *Agrimoniae Eupatoriae* [31]. Zhang et al. separated 8 flavonoids and 5 astragalosides from *Astragali Radix* using 80 min gradient elution with UV and CAD detections connected in tandem [51]. The UV detection was up to 10-fold more sensitive than CAD for the determination of flavonoids. Astragalosides could be quantified only using CAD since no chromophore groups are present in their structure.

#### 2.3.3. Evaluation of Antioxidant Activity

Evaluation of antioxidant activities of apple extracts was determined via detecting oxidation properties obtained using the CD with potentials ranging from 200 to 900 mV at eight electrodes. The sum of peak areas at each detection potential was calculated and plotted as a function of increasing detection potential leading to hydrodynamic voltammograms for each sample. Despite the multi-component character of extracts, preferred oxidation at specific potentials was identified as inflection points (IP) where the curvature of the plot changed. The extracts were divided into three groups shown in Figure 1 based on potentials at which the first IP was observed. The plots featured two IP. First at a potential of 400, 500, or 600 mV, and second at a potential of 700 or 800 mV. This indicated that the oxidation process of compounds presented in the apple extract comprised at least two-electron transfer processes. The first electron transfer at 400–600 mV represented strong antioxidants that donated electrons easily. The second electron transfer at 700–800 mV then characterized oxidation of substances with reduced antioxidant activity. We used signals included in the first electron transfer to compare extracts with respect to the strength of their antioxidants.

For each cultivar, we summed peak areas included in the process of first electron transfer (IP1) (Figure 1D). The results confirmed significant up to a four-fold large difference between cultivars. Moreover, the apparent order of cultivars regarding antioxidant properties was different when the sum of peak areas at all detection potentials was considered (Figure 2) as opposed to the situation when only peak areas of IP1 were evaluated (Figure 1D).

#### 2.3.4. Comparison of Content of Phenolic Compounds and Total Antioxidant Activity of Cultivars

We calculated the Pearson *r* coefficients to compare the detection techniques used. Coefficients were calculated for two different data sets. The first dataset included results of all measured apple cultivars. The second dataset then included five selected apple cultivars that passed the t-test for the similarity between total quantified polyphenols and antioxidant activity. The graphical illustration of results including Pearson *r* values is presented in Figure 3.

The calculated Pearson *r* values ranged from 0.7078 to 0.8337 when all cultivars were evaluated. For a comparison of DAD and CAD, the correlation was estimated to *r* = 0.8337, *p* < 0.0001 with a 95% confidence interval (0.738 to 0.896). The obtained values proved that the DAD and CAD methods were not equal while comparing the dataset of all cultivars.

On the other hand, Pearson *r* coefficients calculated from results obtained for five selected cultivars differed. The *r* values ranged from 0.7132 to 0.9891. The Pearson *r* coefficient calculated for comparison of DAD and CAD was 0.9891, *p* < 0.00001 with a 95% confidence interval (0.9702 to 0.9960). Based on this finding we conclude that the DAD and CAD methods are strongly correlated for overall quantified polyphenols in cultivars ‘Angold’, ‘Golden Delicious’, ‘Meteor’, ‘Reluga’, and ‘Rubinola’.

Correlation coefficients calculated for DAD/CD and CAD/CD comparison of all cultivars were 0.7184 or 0.7078, respectively, while those for selected cultivars were 0.7132 or 0.7867, respectively. For both groups, all and selected cultivars, the coefficients were very similar. Additionally, the scatter of datapoints in Figure 3 for comparison of DAD/CD and CAD/CD in the group of selected cultivars is very similar. Here, unlike the comparison of DAD/CAD for selected cultivars, no improvements were achieved in the group of selected cultivars comparison. We explain that by the difference in working principle of the compared detectors. Unlike the DAD or CAD, where peak area is defined only by the concentration of analyte, the peak area in CD is defined by both concentration and oxidation properties of the analyte. Thus, individual compounds present in the sample even at the same concentration can significantly differ in their oxidation properties and nonequality of results is then expected in most cases.

Based on the evaluation of DAD, CAD, and CD detection techniques we confirmed that CD has a unique position with respect to the evaluation of antioxidant properties of different plants or plant varieties/cultivars. It enables evaluation of all compounds exhibiting the antioxidant properties regardless of their identification. Although DAD and CAD can in some cases provide comparable results, DAD is superior in offering much better accuracy and sensitivity. CAD included in rapid analyses was more prone to overestimation or underestimation of compounds due to the presence of interfering components as discussed in Section 2.3.2. Thus, the combination of DAD and CD represents the best approach to obtain complex information regarding the composition and quantity of analytes and antioxidant properties in the sample.

## 3. Materials and Methods

### 3.1. Chemicals and Solutions

Commercially available standards of phenolic compounds including gallic acid (97.5–102.5%), chlorogenic acid (≥95%), (-)epicatechin (≥90%), rutin hydrate (≥94%), quercetin (≥95%), phloridzin (≥99%), and phloretin (≥99%), HPLC-grade acetonitrile, and methanol used as the mobile phases and extraction solvents, acetic acid (≥99%), formic acid (≥95%), and sodium acetate were all purchased from Sigma Aldrich (Prague, Czech Republic). Unless indicated differently, all reagents were of analytical grade quality. Nitrogen 5.0 (99.999%) for the CAD was obtained from Linde Gas (Prague, Czech Republic).

Standard stock solutions containing 0.5 mg/mL gallic acid, rutin, phloridzin, quercetin, and phloretin, 2 mg/mL chlorogenic acid, and 1 mg/mL epicatechin were individually prepared by dilution in extraction solution. Concentrations of the last two compounds were higher because their higher levels were expected in apple extracts. All stock solutions were stored in the dark at 4 °C. The mixed standard solution contained 5 µg/mL gallic acid, rutin, phloridzin, quercetin, and phloretin, 20 µg/mL chlorogenic acid, and 10 µg/mL epicatechin. These concentrations were selected with respect to the contents in analyzed apple extracts. This mixed standard solution was used for method optimization.

A mixture of 0.1% *v*/*v* acetic acid in methanol was used for extraction of analytes from apples and dilution of analytical standards. Addition of acetic acid was needed to keep the stability of phenolic compounds in solutions and apple extracts [52]. Extraction solutions with addition of standards of seven phenolic compounds at concentration levels of 0.2, 1, 5, 10, 15, and 20 µg/mL were used for evaluation of accuracy.

Sodium acetate buffer for CD was prepared by weighting the respective amount of the compound to obtain a concentration of 10 mmol/L. This solution was acidified with formic acid to pH 3.0 and filtered through a 0.22 µm PTFE membrane filter from Chromservis (Prague, Czech Republic).

### 3.2. Apples and Their Preparation for Analysis

Sixteen traditional apple cultivars explored in this study originated from the experimental plantation of the Research and Breeding Institute of Pomology in Holovousy, Czech Republic. Apples were harvested at optimum harvest maturity in the period from September 11 to October 18, 2018. Cultivars of commercially available apples ‘Santana’, ‘Angold’, ‘Artiga’, ‘Golden Delicious’, ‘Lady Silvia’, ‘Melrose’, ‘Meteor’, ‘Red Jonaprince’, ‘Reluga’, ‘Rubinstep’, ‘Topaz’, ‘Benet’, ‘Golida’, ‘Jarka’, ‘Resista’, and ‘Rubinola’ were analyzed to cover the typical heterogeneity of apples.

Fresh apples after harvesting were continuously extracted according to procedure optimized elsewhere [49]. Our technique included homogenization of 3–5 fruit with peel using a powerful table-top kitchen homogenizer Sencor (Prague, Czech Republic). The homogenates were weighed (3 g), transferred to a 50 mL centrifuge tube, and 15 mL of extraction solution was added. The tube was placed in the ultrasound-assisted extraction bath for 10 min. The sample was centrifuged at 4400 G for 10 min at 4 °C to prevent degradation of analytes. Finally, the supernatant was filtered through a 0.45 µm PTFE syringe filter (Chromservis, Prague, Czech Republic). All extracts were prepared in triplicate and stored in a freezer at −18 °C until analysis.

### 3.3. Chromatography Equipment, Detectors, and Columns

The HPLC/UHPLC system was the Dionex UltiMateTM 3000 RSLC comprising a binary pump, an autosampler, a column oven, a DAD UltiMate 3000 RS, and a CAD Corona Ultra ESA all controlled and evaluated by the ChromeleonTM 7.2 Chromatography Data System (all Thermo Fisher Scientific, Waltham, MA, USA). Since the DAD and CAD were part of the single chromatographic system, detectors were connected in tandem to minimize the number of analyses. These detectors were connected via Viper SST capillary 0.1 × 450 mm (ID × L). Following tandem layout was used: 1. DAD (non-destructive analysis) and 2. CAD (destructive analysis).

Eight-channel coulometric detector CoulArray 5600A ESA (Chelmsford, MA, USA) was used for detection of electroactive substances. This detector was part of the second HPLC/UHPLC platform LC Agilent 1260 Infinity (Santa Clara, CA, USA) that included a quaternary pump and an autosampler. CoulArray 3.10 software was used for data evaluation (Chelmsford, MA, USA).

The fully porous reversed-phase Luna Omega Polar C18 (150 × 4.6 mm; 5 µm) column with polar modification of stationary phase was used for the separations of phenolic compounds. The column was preceded by a guard column Ascentis Express C18 (5 × 4.6 mm) packed with fused core 5 µm particles.

### 3.4. HPLC Separation Using DAD and CAD

The gradient of the mobile phases composed of aqueous acetic acid with pH 2.8 (A) and acetonitrile (B) was used for the chromatographic separation of selected phenolic compounds. The gradient profile was 10–50% B in 10 min, 50% B for 0.2 min, 50–10% B in 2.3 min. The temperature of the column was 30 °C, injection volume 10 µL, and flow rate 1.0 mL/min. The analytes were detected by DAD at a wavelength of 280 (gallic acid, chlorogenic acid, epicatechin, rutin, phloridzin, phloretin), 320 (chlorogenic acid), and 365 nm (rutin, quercetin). CAD evaporating temperature was set at 25 °C, and the frequency of datapoint collection was 20 Hz. Identification of phenolic compounds was achieved by comparison of their retention times with those of standards. Concentrations of phenolic compounds in apple extract were calculated from the integrated peak areas of the identified analytes. The optimal chromatographic separation acquired by DAD and CAD is presented in Appendix A.

### 3.5. HPLC Separation with Coulometric Detection

The mobile phase composition was modified to fit the conductivity detection. Luna Omega Polar C18 and a guard column Ascentis Express C18 were used again for gradient elution with the mobile phases comprising 10 mmol/L sodium acetate buffer acidified to pH 3.0 with formic acid (A) and acetonitrile (B). The gradient profile was 5–30% B in 15 min, 30% B 10 min, 30–50% B in 2 min, 50–5% B in 0.5 min at a flow rate of 1 mL/min. A detector zeroing step (30 s) was included at the beginning of the gradient time and a cell cleaning step (20 s) at the end of each analysis. The temperature of the column was 35 °C, injection volume 10 µL, and flow rate 1 mL/min. Potentials of detection electrodes were set in a range of 200 to 900 mV with 100 mV increments. Peak areas at single detection potentials were summed up to enable comparison of extracts concerning the composition of weak and potent antioxidants.

### 3.6. High-Resolution Mass Spectrometry

The LC analysis with HRMS corroborated characterization of compounds in apple extracts quantified by DAD and CAD, including chlorogenic acid, epicatechin, rutin, phloridzin, and quercetin. Details regarding instrumentation, assay conditions, chromatograms, and MS/MS scans are presented in Appendix A.

## 4. Conclusions

A single detection method such as DAD and CAD is unlikely to be sufficient for the evaluation of phenolic compounds. Each of these methods identifies typical dominant phenolics but does not reflect their antioxidation strength. Therefore, their reducing properties should also be examined to evaluate the antioxidation properties and amounts of electroactive substances via electrochemical detection such as CD. For comparison of DAD and CAD, the emphasis was put on method selectivity and sensitivity to obtain accurate results. The correlation between DAD and CAD was often affected by the presence of interfering substances. Thus, CAD detection was prone to overestimation of analytes due to universal type of response, presence of interfering components, and absence of spectral data. Moreover, analytes present in low quantities suffered from underestimation due to worsened sensitivity of detection compared to DAD. Therefore, extended time gradients and columns with high efficiency had to be applied for analyses of the apple extracts. In contrast, DAD allowed selective and sensitive evaluation of all seven phenolics. Additional information inferred from extracted peak spectra enabled the revelation of co-elutions and eliminate possible errors. Even though the overall time used for the chromatographic separation was mere 12.5 min, all analytes of interest were well separated. Hyphenation of DAD with CD was beneficial since DAD provided complex information concerning identified and quantified main phenolic substances while potential antioxidant activity of compounds in the apple extracts was determined by CD. Separation method using detection techniques described in this study can be used in future research focusing on bioactive compounds present in complex plant samples. However, many more repetitions even in different laboratories will be necessary to evaluate reproducibility and repeatability of our method and to make it an official method.

## Figures and Tables

**Figure 1 molecules-26-03246-f001:**
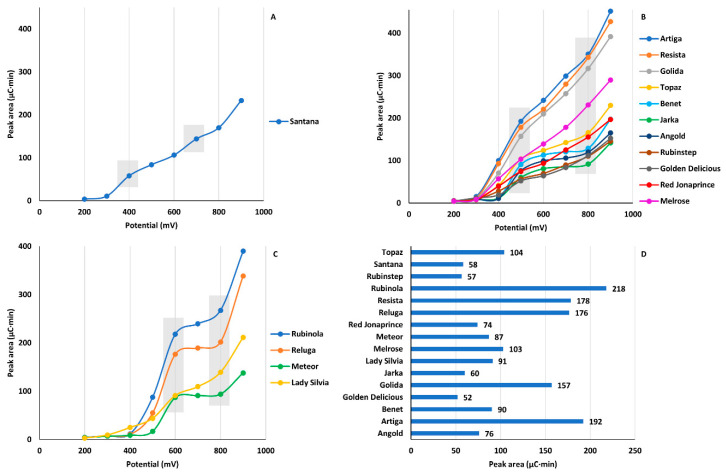
Evaluation of antioxidant activity using CD. Cumulative peak areas at each detection potential are plotted as a function of increasing detection potentials to a hydrodynamic voltammogram. Samples are divided by applied potential at which first inflection point (IP, meaning change of waveform direction) occurs, at 400 (**A**), 500 (**B**), and 600 mV (**C**). Comparison of samples based on sum of peak areas included in first electron transfer (IP 1) (**D**).

**Figure 2 molecules-26-03246-f002:**
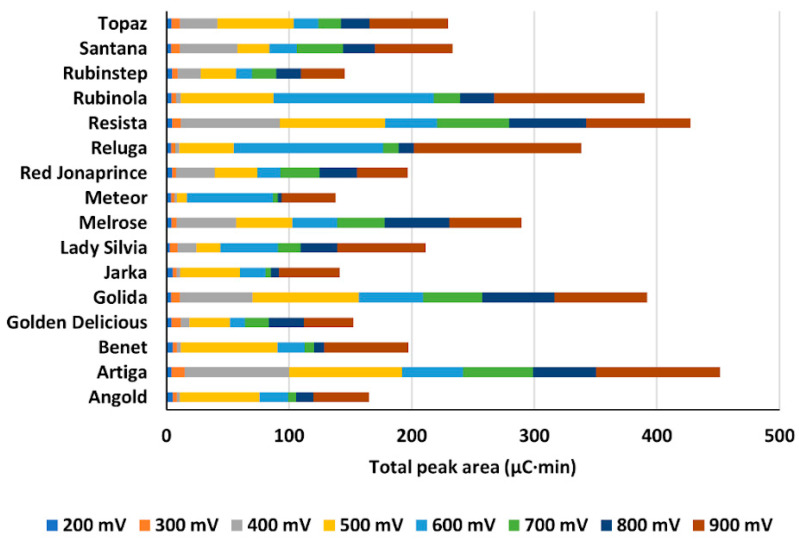
Contributions to total peak area obtained using CD at single potentials ranging from 200–900 mV.

**Figure 3 molecules-26-03246-f003:**
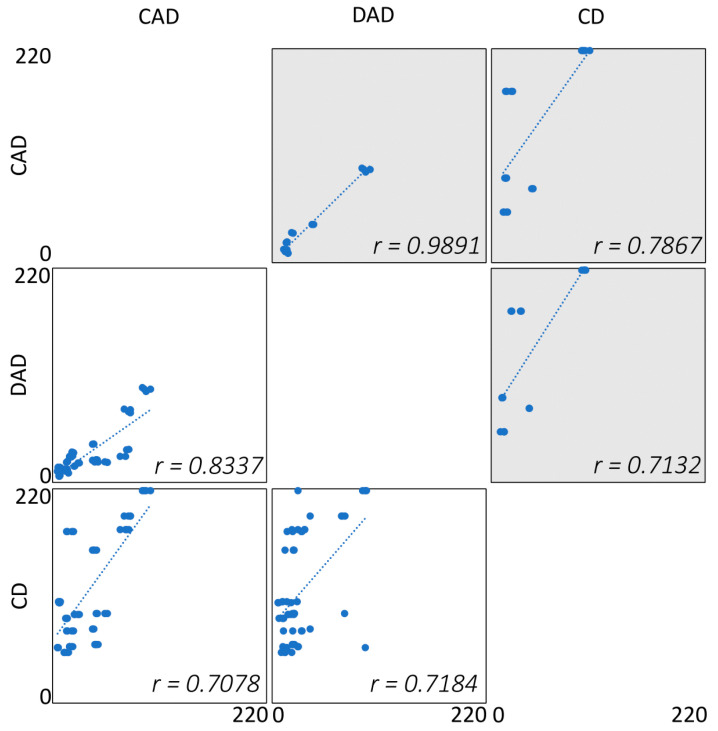
Pearson *r* correlation between detectors using all cultivars (62 data points, white background) and selected cultivars that passed the *t*-test for similarity between total quantified polyphenols and antioxidant activity (18 data points, light grey background).

**Table 1 molecules-26-03246-t001:** System suitability test (n = 6) of the elution of phenolic compounds detected using diode-array (DAD) and charged aerosol (CAD). Separation conditions: gradient elution with mobile phases aqueous acetic acid with pH 2.8 (A) and acetonitrile (B), gradient profile 10–50% B in 10 min, 50% B 0.2 min, 50–10% B in 2.3 min, column temperature 30 °C, injection volume 10 µL, flow rate 1.0 mL/min. Concentration of standard mixture 5 µg/mL.

DAD Detector	t_R_ ^1^ (min)	Repeatability, RSD (%)		S ^3^	R_S_ ^4^	P_C_ ^5^	R_f_ ^6^
		t_R_	A ^2^				
Gallic acid 280 nm	2.81	0.15	0.19	0.91	-	49.77	0.63
Chlorogenic acid 320 nm	4.47	0.14	0.36	0.95	17.55	58.60	1.6
Epicatechin 280 nm	5.19	0.17	0.50	0.82	7.12	54.61	2.02
Rutin 254 nm	6.09	0.15	0.16	0.97	9.36	67.78	2.54
Phloridzin 280 nm	7.60	0.11	0.16	0.87	15.50	45.72	3.42
Quercetin 365 nm	9.57	0.11	0.51	1.20	17.93	27.61	4.56
Phloretin 280 nm	10.48	0.10	0.52	0.91	8.23	43.67	5.09
**CAD Detector**							
Gallic acid	2.83	0.65	0.95	0.96	-	79.41	0.65
Chlorogenic acid	4.50	0.42	0.40	0.93	17.53	79.41	1.62
Epicatechin	5.22	0.33	0.36	0.86	7.16	64.29	2.04
Rutin	6.12	0.25	0.42	0.95	9.38	67.50	2.56
Phloridzin	7.64	0.20	0.67	0.87	15.45	61.36	3.44
Quercetin	9.60	0.20	0.39	1.16	17.91	64.29	4.58
Phloretin	10.51	0.16	0.09	0.91	8.27	61.36	5.11

^1^ Retention time; ^2^ Area; ^3^ Symmetry factor; ^4^ Resolution of peaks; ^5^ Peak capacity; ^6^ Retention factor.

**Table 2 molecules-26-03246-t002:** Limits of detection and quantitation, calibration ranges, regression equations, and determination coefficients using diode-array (DAD) and charged aerosol (CAD) detectors. Separation conditions: gradient elution with mobile phases aqueous acetic acid with pH 2.8 (A) and acetonitrile (B), gradient profile 10–50% B in 10 min, 50% B 0.2 min, 50–10% B in 2.3 min, column temperature 30 °C, injection volume 10 µL, flow rate 1.0 mL/min.

DAD Detector	LOD (µg/mL)	LOQ (µg/mL)	Calibration Range (µg/mL)	Regression Equation	R^2^
Gallic acid 280 nm	0.03	0.10	0.10–20	0.2393x − 0.0132	0.9989
Chlorogenic acid 320 nm	0.03	0.10	0.10–20	0.2547x − 0.0298	0.9983
Epicatechin 280 nm	0.07	0.25	0.25–20	0.0876x − 0.0083	0.9985
Rutin 365 nm	0.07	0.25	0.25–20	0.1584x − 0.0092	0.9990
Phloridzin 280 nm	0.03	0.10	0.10–20	0.2174x − 0.0100	0.9992
Quercetin 365 nm	0.03	0.10	0.10–20	0.3971x − 0.0820	0.9977
Phloretin 280 nm	0.03	0.10	0.10–20	0.3791x − 0.0162	0.9992
**CAD Detector**					
Gallic acid	0.30	1.00	1.00–20	0.0058x − 0.0024	0.9949
Chlorogenic acid	0.30	1.00	1.00–20	0.0096x − 0.0031	0.9943
Epicatechin	0.30	1.00	1.00–20	0.0163x − 0.0047	0.9973
Rutin	0.30	1.00	1.00–20	0.0145x − 0.0017	0.9985
Phloridzin	0.30	1.00	1.00–20	0.0196x − 0.0019	0.9995
Quercetin	0.30	1.00	1.00–20	0.0224x − 0.0116	0.9956
Phloretin	0.15	0.50	0.50–20	0.0309x − 0.0420	0.9996

R^2^ coefficient of determination.

**Table 3 molecules-26-03246-t003:** The accuracy of method expressed in terms of compound recovery at six concentration levels using samples Scheme 0.2; 1; 5; 10; 15 and 20 µg/mL), and intra-day precision. Separation conditions: gradient elution with mobile phases aqueous acetic acid with pH 2.8 (A) and acetonitrile (B), gradient profile 10–50% B in 10 min, 50% B 0.2 min, 50–10% B in 2.3 min, column temperature 30 °C, injection volume 10 µL, flow rate 1.0 mL/min.

Analyte/Spiked Level (µg/mL)	Recovery (%)						Intra-Day Precision (%)
	DAD						
	0.2	1	5	10	15	20	Cultivar HL 1343
Gallic acid	81.7	98.5	94.3	93.7	93.6	89.3	
Chlorogenic acid	93.2	101.6	96.3	97.4	89.6	86.8	3.4
Epicatechin	86.0	95.4	97.6	100.2	97.5	90.9	7.1
Rutin	94.6	104.8	100.1	102.7	97.0	92.8	3.2
Phloridzin	100.3	97.4	92.8	94.3	93.1	88.0	1.9
Quercetin	85.2	82.9	86.2	81.1	82.2	81.0	
Phloretin	85.1	89.4	86.8	89.2	90.0	86.6	
	**CAD**						
Gallic acid	100.9	99.7	102.0	105.8	102.3	101.4	
Chlorogenic acid	108.2	109.7	109.4	106.0	97.0	104.1	5.1
Epicatechin	114.3	116.2	117.5	108.7	98.6	103.2	7.3
Rutin	93.5	97.8	98.7	90.6	92.1	92.4	7.8
Phloridzin	88.6	97.0	89.1	92.8	88.2	88.8	6.0
Quercetin	80.8	87.7	88.3	89.7	83.6	86.6	
Phloretin	87.2	90.2	88.5	87.0	88.1	86.9	

**Table 4 molecules-26-03246-t004:** Statistics for comparison of sensitivity of diode-array detector and charged aerosol detector based on calibration slopes.

Compound	*p*-Value
Gallic acid	7.24 × 10^−6^
Chlorogenic acid	2.67 × 10^−6^
Epicatechin	2.58 × 10^−5^
Rutin	6.12 × 10^−6^
Phloridzin	1.45 × 10^−7^
Quercetin	1.81 × 10^−7^
Phloretin	3.05 × 10^−7^

Significance level α = 0.05.

## Data Availability

The data presented in this study are available in this article and in the Appendix A.

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
