# Peer review of "Benefits and Pitfalls of HPLC Coupled to Diode-Array, Charged Aerosol, and Coulometric Detections: Effect of Detection on Screening of Bioactive Compounds in Apples"

_molecules, 2021, doi:10.3390/molecules26113246_

Round 1

Reviewer 1 Report

The results are very well presented and analyzed. However, for it to be an official method, many more repetitions will be necessary, even in different laboratories to be able to evaluate the reproducibility of the methodology.

Author Response

As suggested by the reviewer the additional information was added. We agree that the presented method requires more analyses even in different laboratories to be considered as an official. The sentence was added in the conclusion (lines 476-478) for better clarification of the present method status.

Reviewer 2 Report

I do not commend this manuscript to be published in Molecules as it is highly similar to the paper [29]. Only changes in apple samples and chromatographic conditions could not be regarded as a novelty as both studies are based on the comparison with the DAD, ECD, and CAD detectors.

[29] Plaza, M; Kariuki, J; Turner, C. Quantification of Individual Phenolic Compounds' Contribution to Antioxidant Capacity in Apple: A Novel Analytical Tool Based on Liquid Chromatography with Diode Array, Electrochemical, and Charged Aerosol Detection. JOURNAL OF AGRICULTURAL AND FOOD CHEMISTRY 2014, 62(2), 409-418. (DOI: 10.1021/jf404263k)

Author Response

We included this report in the Introduction of the manuscript since it fits our scope from several points of view. Indeed, it presents combination of DAD and universal CAD detection in tandem, adds additional information regarding quantity of the respective analytes, and assessment of antioxidant properties using cyclic voltammetry. Obviously, we could not omit this reference. However, the aims of their study differ considerably from those presented in our manuscript. The mentioned study was focused on (i) replacing tedious antioxidant assays using free radical molecules with cyclic voltammetry method and (ii) using CAD to quantitate some compounds using only a few standards (when standards are not available).

In contrast, we compared DAD and CAD regarding their selectivity and sensitivity when complex matrices are analysed, i.e. an issue that was not included in ref. 29. Moreover, their chromatographic separation lasting for more than 50 min is less suitable for rapid screening.

We added a sentence in the text emphasizing that to enhance the selectivity of analyses, the methods often suffered from long separation times (lines 90-91).

Reviewer 3 Report

The authors presented method for determination of major phenolic compounds in apples. In my opinion, the analytical techniques used in this study are not innovative, but the article is nevertheless quite carefully written.

The manuscript (ID: molecules-1208483) entitled “Matrix effects in HPLC coupled to diode-array, charged aerosol, and coulometric detections: Effect of detection on screening of bioactive compounds in apples” can be recommended for publication in the Molecules in present form .

Author Response

Thank You for manuscript evaluation. We would like to provide a comparison with previously published work, also cited in the manuscript as the reference [29] and others which scope may seem similar to ours and explain differences in the manuscript text.

Reviewer 4 Report

The title does not reflect the content (experimental part and conclusion) of the article. 

As it was correctly mentioned in the introduction, a matrix effect is well known in the field of MS (selective detector), where matrix effect takes place during ionization process and alters the accuracy of measured concentrations. Using less selective detectors, it is normal that different interferences occur during detection process. For that reason, LC method must be long enough to separate effectively all compounds of interest. 

Results and Discussion Section is devoted to the validation of LC-DAD and LC-CAD methods, the comparison of obtained results by both methods, and the evaluation of antioxidant activity of samples using LC-CD method. The comparison of quantified phenolic compounds in apples using LC-DAD/CAD and antioxidant activity of cultivars is also provided.

Lines 345-350 and 441-442 do not reflect the outcomes of the evaluation of detection methods. It is clearly presented that DAD is superior detection system compared to CAD. There is no reason why systems should work in tandem. All investigated phenolic compounds can be detected using simple LC-DAD system with much better accuracy and sensitivity compared to that of CAD system. If authors include in the methods compounds that do not contain chromophore groups and cannot be detected using DAD than CAD can give additional value to the method. But it is not a case in this study.

Authors should change the title that reflects the content of article and rewrite the introduction that is devoted to the problem discussed in results and discussion section where the detection methods LC-DAD and LC-CAD are evaluated and compared. The results obtained during the study are interesting and can be useful if they are presented correctly.

Abstract: CoulArray detection was used for advanced characterization of extracts in terms of antioxidant content and strength to obtain more complex information concerning sample composition. This detection allowed detailed assessment of even unidentified compounds with antioxidant activity.

I didn’t find any proof in the article for that statement. Please delete it.

HPLC/UHPLC separation using diode array combined in tandem with coulometric detections represented the best approach enabling quick, yet complex characterization of bioactive compounds in apples.

The systems are not combined in tandem. They work separately. Please correct the statement accordingly.

Line 227-228. It is not understandable for what reason authors applied LC-HRMS to confirm the presence of the quantified analytes? Please justify or delete that section and corresponding information.

Round 2

Reviewer 2 Report

No comment.

Reviewer 4 Report

No suggestions

This manuscript is a resubmission of an earlier submission. The following is a list of the peer review reports and author responses from that submission.

Round 1

Reviewer 1 Report

As the title says, this manuscript presents detailed analysis of three analytical methods (DAD, CAD and CD) for measuring the concentration and antioxidant activity of polyphenols in apples.

This manuscript presents many defects that prevent its publication in Molecules.

Main concern is the lack of focus. Several aspects are mixed in this study:
- is it a comparison of the three different analytical methods ?
Then, the analytic aim is never mentioned, (is it characterisation of individual polyphenols, estimate of their concentration, dosing of the antioxidant activity, ... )
- is it a study of the cultivar matrix effect on the fidelity of the measure, but this is never estimated.

Another aspect is that, whatever the analytical aim, no clear conclusion is ever drawn.
- Comparison between DAD CAD is only superficial despite the amount of data.
- there is no estimate on the fidelity of the measure with respect to the matrix, as only one measure is mentioned (even though extract are said to be in triplicate)
- correlation between measured total polyphenol and antioxidant activity is mentioned, once as being good (l-287) once as not significant (l-383). But this is hand-waving, as all what we have is a lousy bar plot (fig 3) (where a scatter plot of content vs activity would MUCH clearer) and absolutely NO quantitative analysis (where a simple R or a Pearson of the scatter plot would be a strict minimum)

There is a lot of strange aspects in the analysis which should also be cleared:, to cite the most striking ones:

  • most of all, table S1 is very strange
    • how TOTAL poyphenol is obtained is not described
    • despite very large discrepancies between DAD and CAD, the TOTAL polyphenol comparison is quite good - this is never discussed
    • certain values are given as 0.0 while others as <LOQ (does it means it is >LOD ???)
  • - very strange linearity behaviour, with (if I got it well) so large negative offset that it leads to negative regression values at the LOQ for several compounds (Chlorogenic ac. and quercetin with DAD, )
  • values reported wrongly between SI and main text (e.g. +1174.6% (sic) overestimate (l216) where it is more like infinity to me ! (look at Golden or Meteor))
  • silly analysis of rubinola being in "close match" (l289) whereas it is used as the 100% reference!!!
    - no resolution factor for gallic acid chromato peak

In conclusion, I think this work should not be published.

Author Response

Reviewer1

Comments and Suggestions for Authors

As the title says, this manuscript presents detailed analysis of three analytical methods (DAD, CAD and CD) for measuring the concentration and antioxidant activity of polyphenols in apples.

This manuscript presents many defects that prevent its publication in Molecules.

Main concern is the lack of focus. Several aspects are mixed in this study:
- is it a comparison of the three different analytical methods ?

Comparison was focused on detection techniques instead of analytical methods while quick LC separation is applied in connection with both, DAD and CAD. The aim was added at the end of Introduction part (lines 101-103).

Then, the analytic aim is never mentioned, (is it characterisation of individual polyphenols, estimate of their concentration, dosing of the antioxidant activity, ... )

- is it a study of the cultivar matrix effect on the fidelity of the measure, but this is never estimated.

The aim is mentioned at lines 98-101.

Matrix effects were studied in wide range of cultivars and the main problems are pointed out – e.g. cultivars with higher chlorogenic acid content. This text was added to the lines 103-104.

Another aspect is that, whatever the analytical aim, no clear conclusion is ever drawn.

- Comparison between DAD CAD is only superficial despite the amount of data

Comparison of different detections was focused on data that could be generalized (lines 104-105). Detailed comparison is visible in additional Table S1 in Supplementary material. This Table S1 was completed with SD values that were omitted just to keep table simplicity showing all data. Values in the table were also recalculated with calibrations using all points in triplicate that enable to increase precision of the obtained data (lines 147-149).

- there is no estimate on the fidelity of the measure with respect to the matrix, as only one measure is mentioned (even though extract are said to be in triplicate)

SD values were added to the Table S1 with the phenolic content results. The highest RSD was then applied to calculate SD for total content of polyphenols that corresponds to the sum of the selected main components quantified in apple extracts. Manuscript text on lines 232-237 was modified.

- correlation between measured total polyphenol and antioxidant activity is mentioned, once as being good (l-287) once as not significant (l-383). But this is hand-waving, as all what we have is a lousy bar plot (fig 3) (where a scatter plot of content vs activity would MUCH clearer) and absolutely NO quantitative analysis (where a simple R or a Pearson of the scatter plot would be a strict minimum)

Based on this valuable remark, comparison of methods was reprocessed (lines 333-349). A scatterplot was constructed for the comparison of polyphenols quantified by CAD and DAD, and antioxidant activity evaluated by CD. The Pearson r correlation was estimated to 0.71, p < 0.05 for a DAD detector and 0.72, p < 0.05 for a CAD detector. The similarity of Pearson’s r is not providing a definitive difference of superiority between the two methods overall with a p = 0.482. To estimate the difference, the two-tailed student t-test with a significance of 95% was used between each cultivar to estimate the similarity of total quantified polyphenols and antioxidant activity.

There is a lot of strange aspects in the analysis which should also be cleared:, to cite the most striking ones:

  • most of all, table S1 is very strange

Table S1 was completed with SD values for the respective polyphenols content and for total polyphenols the highest RSD was applied to calculate precision. Manuscript text on lines 232-237 was modified.

  • how TOTAL polyphenol is obtained is not described

Total polyphenol content was based on the sum of the main detected polyphenolic compounds mentioned in the table – this was added to the table title and mentioned in the manuscript text as well (lines 232-237).

  • despite very large discrepancies between DAD and CAD, the TOTAL polyphenol comparison is quite good - this is never discussed

The discussion was modified following this valuable comment (lines 260-265)

Despite the upper mentioned detector-based differences of individual quantified polyphenols, a correlation of total quantified polyphenol content obtained by DAD and CAD detections was found in some cultivars ('Angold', 'Artiga', 'Golden Delicious', 'Meteor', and 'Rubinola'). But it should be pointed out that such correlation can be affected by DAD corresponded to higher chlorogenic acid content and CAD with higher epicatechin. Then, sum of the combination of all analytes can cause similarity of total polyphenols content obtained by both methods.

  • certain values are given as 0.0 while others as <LOQ (does it means it is >LOD ???)

Value 0.0 means that no peak was found in the respective retention time. Values lower than LOD corresponds to lower signal than LOD but with visible signal in the respective retention time. Values lower than LOQ are higher than LOD but are not enough to evaluate the respective concentration. This comment was added to the manuscript text to explain data evaluation properly (lines 234-237).

  • - very strange linearity behaviour, with (if I got it well) so large negative offset that it leads to negative regression values at the LOQ for several compounds (Chlorogenic ac. and quercetin with DAD, )

The obtained results were affected by small relative values of the peak area that are evaluated using the mentioned instrumentation and precision was increased using evaluation of all measures. (revisions in Table 2) that enabled recalculation of values in Table S1. Following EMA guidelines blank and zero samples should not be taken into consideration to calculate the calibration curve parameters. This comment was added to the manuscript text (lines 147-149).

LOQ values were calculated using the signal-to-noise ratio and were found to be very similar in comparison with other LC instrumentation except of higher sensitivity in case of the used Dionex chromatograph were capillary connections were modified to decrease dispersion and thus increase sensitivity. This was already mentioned in the manuscript text (lines 176-181).

  • values reported wrongly between SI and main text (e.g. +1174.6% (sic) overestimate (l216) where it is more like infinity to me ! (look at Golden or Meteor))

Comparing some values that were close to LOQ values using one detection high differences were demonstrated. Thus, one detection can affect the obtained results quite a lot and thus big differences were described to prevent errors in routine analyses. This text was added to the manuscript (lines 256-259).

  • silly analysis of rubinola being in "close match" (l289) whereas it is used as the 100% reference!!!

Thank you for this comment comparison of polyphenolic content and antioxidant activity was reprocessed by different method to enable proper comparison of obtained results. The manuscript text was modified, and only comparable cultivars were mentioned (lines 333-349).

  • - no resolution factor for gallic acid chromato peak

The resolution of chromatographic peaks is calculated between the peak of interest and the closest neighbour peak in the elution order. Thus, in our case, analysis of 7 phenolic compounds is characterised by 6 resolution factors. From the common practice, the first eluting chromatographic peak was not characterised by resolution. In case of resolution of gallic acid, as the first peak calculated from the dead volume, the resolution was always higher than 5. This comment was added to the manuscript text (line 134-136).

  • A close match was found for cultivars 'Rubinola', 'Artiga', 'Angold', and 'Santana'.

In the process of revisions, different kind of data comparison was suggested by reviewer. Thus, comparison of methods was reprocessed (lines 372-377). A scatterplot was constructed for the comparison of polyphenols quantified by CAD and DAD, and antioxidant activity evaluated by CD. The Pearson r correlation was estimated to 0.71, p < 0.05 for a DAD detector and 0.72, p < 0.05 for a CAD detector. The similarity of Pearson’s r is not providing a definitive difference of superiority between the two methods overall with a p = 0.482. To estimate the difference, the two-tailed student t-test with a significance of 95% was used between each cultivar to estimate the similarity of total quantified polyphenols and antioxidant activity. (lines 333-349)

In conclusion, I think this work should not be published.

We hope that all the mentioned comments help to increase the manuscript level and understanding of our aims and results.

Reviewer 2 Report

This study describes the effects of biological matrices on analytical parameters for quantification based on HPLC/UHPLC DAD.

Whilst being of interest to a broad readership, this reviewer would perhaps have suggested transfer to Foods or a more appropriate medium for this manuscript. Regardless, comments for improvement to this well-described study are below:

  • discussion of antioxidants could benefit from additional references described below:

1. https://doi.org/10.3390/molecules25020342

2. https://doi.org/10.1038/s41598-020-65769-5

3. https://doi.org/10.3390/nu12030753

4. https://doi.org/10.3390/nu12040974

  • whilst on line 99 it is reasonable to link to other protocols, in the interest of the readership for such a paper, it would be advisable to describe this here.
  • Line 106 - the adherence to the EMA guideline in ref [46] should be explicitly described to support the viability of the methodology.
  • Table 1: could the authors describe their approach to calculating repeatability, taking into account the uncertainty/bias within and out of batch?
  • Could the authors provide a more substantial commentary of the nature of the variance in Table 2 relating to methodology and analyte, linking to selectivity?
  • Line 283: perhaps I have missed this, however how was the content of substances normalized and by which method? What rationale determined the method?

Author Response

Reviewer 2

Comments and Suggestions for Authors

This study describes the effects of biological matrices on analytical parameters for quantification based on HPLC/UHPLC DAD.

Whilst being of interest to a broad readership, this reviewer would perhaps have suggested transfer to Foods or a more appropriate medium for this manuscript. Regardless, comments for improvement to this well-described study are below:

  • discussion of antioxidants could benefit from additional references described below:
  1. https://doi.org/10.3390/molecules25020342
  2. https://doi.org/10.1038/s41598-020-65769-5
  3. https://doi.org/10.3390/nu12030753
  4. https://doi.org/10.3390/nu12040974

The discussion was completed using the up-to-date proposed sources to complete the part connected with antioxidant properties that can be analysed different ways – using spectrophotometric methods, combining spectrophotometry with chromatography (LC prior and after DPPH) or application of NMR. The discussion was modified (lines 92-97).

“For in vitro spectrophotometric evaluation of antioxidant activity, also use of ABTS, DPPH, FRAP, and ORAC tests is commonly accepted [43-45]. By chromatographic analysis of plant extract before and after treatment with reactive components of named tests, the major components with strong antioxidant activity can be easily identified [43]. For additional nutritional characterization of samples and structural characterization of isolated compounds nuclear magnetic resonance can be employed [43-46].”

  • whilst on line 99 it is reasonable to link to other protocols, in the interest of the readership for such a paper, it would be advisable to describe this here.

The previous LC method was based on application of core shell C18 stationary phase which was changed to fully porous C18 with modification for higher resolution of polar substances. Then, gradient profile was modified to increase resolution of substances close to chlorogenic acid. The comment was added to the manuscript text (lines 108-111).

  • Line 106 - the adherence to the EMA guideline in ref [46] should be explicitly described to support the viability of the methodology.

EMA is recommended for validation of bioanalytical methods. The EMA guidelines are very close to FDA that is intended for application to food matrix but more specific in some validation parameters and thus were applied to this work. Additionally, EMA validation guideline well describes cross validation of method, what is our case. This comment was added to manuscript text to clearly describe our choice (lines 118-121).

  • Table 1: could the authors describe their approach to calculating repeatability, taking into account the uncertainty/bias within and out of batch?

Our results included in Table 1 corresponds to standard repeatability evaluated in frames of system suitability test common in LC separations. Values in the last column of Table 3 demonstrate in batch sample repeatability (precision) and Table S1 completed with SD values shows variability of all samples but again within the batch. The comment was added to the manuscript text (lines 130-132).

  • Could the authors provide a more substantial commentary of the nature of the variance in Table 2 relating to methodology and analyte, linking to selectivity?

The variance between detectors is primary caused by different operating principle of detectors – universal CAD and specific DAD (detection of substances containing chromophores enabling absorption of spectral wavelengths). The variance within detector can be caused by different structures of analytes – different ability to absorb certain spectral wavelength (DAD), formation of non-volatile analytes after evaporization of mobile phase (CAD). The selectivity is again increased in case of DAD while peak purity can be evaluated. In case of CAD co-elution with other substances cannot be visualized. The comment was added to manuscript text (lines 157-163).

  • Line 283: perhaps I have missed this, however how was the content of substances normalised and by which method? What rationale determined the method?

In the process of revisions, different kind of data comparison was suggested by reviewer. Thus, comparison of methods was reprocessed (lines 333-349). A scatterplot was constructed for the comparison of polyphenols quantified by CAD and DAD, and antioxidant activity evaluated by CD. The Pearson r correlation was estimated to 0.71, p < 0.05 for a DAD detector and 0.72, p < 0.05 for a CAD detector. The similarity of Pearson’s r is not providing a definitive difference of superiority between the two methods overall with a p = 0.482. To estimate the difference, the two-tailed student t-test with a significance of 95% was used between each cultivar to estimate the similarity of total quantified polyphenols and antioxidant activity.

Round 2

Reviewer 1 Report

In my first review I made strong criticisms on the lack of a clear statement on the aims and goals of this work.
This has somehow been corrected in the present version, even though I still think there is a lack of a clear focus in this study.
Some other points have been corrected by the authors, such as the evaluation of the fidelity of the measurements which is now somehow presented as SD in table S1.

There are however additions which are VERY problematic.

1/ a correlation between antioxydant activity and total polyphenolic content measured either by CAD or DAD is now supposedly presented in a scatter plot with Pearson R values.
BUT the plot does not make sense - the captions says that CAD or DAD amount is on x axis while CD is on y, nevertheless each cultivar, for which only one CD value should be given do not show up on a single horizontal line (one CD value) but seems to be on 2 different (often very different) values. ???
Then a more careful analysis seems to show some symmetry, where, for each cultivar, the CD values for the CAD points is the identical to the DAD values ???
So is it just a fake, an error, a wrong caption, or just BS ?
Finally, mentioning a p value of 0.482 (sic !) seems to be the sign of authors not knowing what they are doing.

2/ was added L234-237:
"Value 0.0 means that no peak was found in the respective retention time. Values lower than LOD corresponds to lower signal than LOD but with visible signal in the respective retention time. Values lower than LOQ are higher than LOD but are not enough to evaluate the respective concentration
Does not make sense - "lower than LOD" means it is not detectable - so "but with visible signal" is not relevant
"lower than LOQ" means it is not quantifiable reliably - so no value value should be given.
Anything more is just meaningless.

Moreover, some the other points which were a source of criticisms are still problematic, in particular all my remarks on the values in table S1 and the way it is reported in the text, remain.

In conclusion, I still advise rejection of this manuscript in the present form.